# The CLASS (Cerebral visual impairment Learning and Awareness for School Staff) Pilot Study: An evaluation of the awareness of CVI amongst teachers and comparative evaluation of two different educational resources on understanding

**Aloka Jayasinghe[1‡], Helen St Clair Tracy[2‡*], John Ravenscroft[3], Andrew Blaikie[2]**

**1** School of Medicine, University of St Andrews, St Andrews, United Kingdom, **2** Infection & Global Health Division, School of Medicine, University of St Andrews, St Andrews, United Kingdom, **3** Moray House School of Education & Sport, University of Edinburgh, Edinburgh, United Kingdom

‡ These authors are joint first authors.
* hsct1@st-andrews.ac.uk

## Abstract

Cerebral visual impairment (CVI) is the leading cause of visual impairment in children in high income countries. Despite its prevalence, awareness of CVI among educators remains low, meaning that many affected children may not receive the support they need in school. While previous research has highlighted the challenges faced by children with CVI, few studies have systematically assessed teacher awareness and the effectiveness of targeted educational interventions in improving classroom practices. This study addresses this gap by evaluating: (1) teacher awareness of CVI, (2) existing classroom practices that may impact children with CVI, (3) the effectiveness of two CVI educational media formats (video and text) in increasing understanding, and (4) the changes teachers would be willing to implement following exposure to these resources. By comparing the impact of these two formats, this study provides insights into how best to deliver CVI training for teachers in a way that is both accessible and effective. A total of 111 teachers from primary, secondary, and special schools across the UK participated in a survey incorporating either a three-minute video simulation or a 1.5-minute text-based resource about CVI. Before exposure, 72% of participants had not heard of CVI, with awareness particularly low among mainstream teachers (98% of primary and 80% of secondary teachers were unaware**). Teachers also reported inconsistent use of CVI-supportive practices, such as reducing classroom clutter and simplifying smart screen content. Both media formats significantly increased teachers' willingness to implement changes ($p < 0.0001$). The text format showed a slightly greater increase in average Likert scores, and the Wilcoxon signed-rank test revealed a larger statistical effect for text ($z = -12.91$) compared to video ($z = -8.90$). However, the video format was also highly effective, producing a similarly

**Data availability statement:** All relevant data are within the paper and its Supporting information files (S3 Appendix contains the anonymised dataset used for analysis).

**Funding:** This research was supported by a Laidlaw Scholarship at the University of St Andrews. The funder had no role in study design, data collection and analysis, decision to publish, or preparation of the manuscript.

**Competing interests:** The authors declare that they have no competing interests.

strong impact, with both formats achieving an identical median increase of 1.0. These results suggest that while text may have led to slightly larger shifts in rank-based scores, the video format remained a powerful and engaging tool for increasing teachers' willingness to implement CVI-supportive strategies. The findings suggest that small, manageable adaptations, such as reducing visual distractions and maintaining consistency in classroom layouts, are practical for teachers and may have a meaningful impact on children with CVI. This study highlights the potential of bite-size learning resources in raising awareness and encouraging evidence-based teaching adaptations. By providing concise, accessible materials, teachers can be equipped with strategies to support children with CVI while minimising additional workload demands. Future efforts should focus on scaling these resources to reach a wider audience, including families and caregivers, to foster a more inclusive understanding and response to CVI.

## Introduction

The widely cited claim that "80% of learning is related to vision" [1,2] oversimplifies the complex relationship between vision and learning. While the exact origin of this statistic remains unclear, research consistently demonstrates the crucial role of vision in learning and development, particularly in reading, writing, spatial reasoning, and interpreting visual stimuli. Foundational work by Hubel and Wiesel [3] highlighted the importance of visual input for brain development, showing through animal studies that the visual cortex develops in response to environmental stimuli, particularly during critical periods. Subsequent research has extended these findings to humans, underscoring the necessity of early visual experiences for typical brain development [4]. These studies provide a scientific basis for early interventions in cases of visual impairment, ensuring optimal neural development.

When learning does not develop typically, as in cases of children with learning difficulties, developmental delays, or intellectual disabilities, vision-related challenges may be both a contributing factor and an unmet need. Research indicates that children in special education settings are disproportionately affected by undiagnosed or uncorrected vision problems [5]. Black [5] conducted comprehensive in-school eye examinations with 200 children in a UK special education school, finding that 43% required spectacles. The study implemented interventions such as prescribed spectacles, patching therapy for amblyopia, and ophthalmic referrals, leading to significant benefits in academic engagement, quality of life, behaviour, and educational accessibility. The same study found that 23.5% of children exhibited visual perception difficulties, suggesting potential unmet needs related to brain-based visual processing difficulties. While more research is needed, these findings strongly suggest that vision plays a role in learning challenges, even if the precise extent remains unclear.

Brain-based visual processing difficulties are commonly referred to as cerebral visual impairments (CVI), sometimes called cortical visual impairment. CVI is an umbrella term encompassing one or more atypical visual processes in the brain

[6–8]. Vision serves multiple functions, including guiding movement, enabling social interaction, accessing information, and promoting learning [9]. Approximately 40% of the brain is involved in processing vision [10], with intricate relationships between vision and cognitive functions, making it difficult to isolate specific contributions [11]. However, distinct visual functions such as perceiving detail, colour, motion, spatial relationships, and object recognition are essential for navigating environments, fostering meaningful connections, and acquiring knowledge [12]. In educational settings, unmet visual needs can significantly impact learning and classroom engagement.

Low awareness of CVI among teachers can have important social implications. Without a clear understanding of the condition, children's behaviours may be misinterpreted, for example as inattentiveness or disengagement, rather than recognised as indicators of unmet access needs. These misunderstandings can unintentionally lead to reduced support, missed learning opportunities, and social exclusion, particularly in mainstream education settings, where teachers are already navigating a wide range of learning needs.

The definition of CVI remains a topic of ongoing debate. A recent U.S. initiative proposed five key elements to define CVI, emphasising brain-based visual impairments that exceed what would be expected from ocular conditions alone [13]. This definition incorporates both lower-order deficits, such as visual acuity, and higher-order impairments, such as object recognition and spatial processing. However, it could be criticised for excluding cases of CVI acquired in adulthood, such as post-stroke visual impairment [14], despite similar underlying neurological mechanisms between CVI acquired in childhood and adulthood. By contrast, an earlier definition framed CVI as "a verifiable visual dysfunction which cannot be attributed to disorders of the anterior visual pathways or any potentially co-occurring ocular impairment" [15]. While useful as a clinical starting point, this definition lacked guidance on the functional characteristics of CVI, particularly its broader impact in educational and developmental contexts [16].

The spectrum of special needs among children is vast and spans both special and mainstream schools. Williams [17] reported that 57% of children in a special school had visual difficulties likely due to CVI, while 35% of children (73 of 207) with additional support needs in mainstream schools also exhibited CVI-related visual difficulties within the same study. Among all mainstream school children, including those with additional support needs, Williams found that 3.4% were affected by CVI. Notably, among those tested without identified additional support needs, 5 out of 41 were found to have visual difficulties related to CVI. Despite these findings, the true prevalence of CVI remains unclear, necessitating further research to better understand its scope and impact.

In Scotland, where approximately one million children reside, 889 have a formal CVI diagnosis, accounting for 57% of all children notified to the register. However, this represents only 0.09% of the childhood population, while the prevalence estimates from Williams [17] suggest that at least 3.4% (34,000 children) are affected. This means only 2.6% of children likely to have CVI have been formally diagnosed, leaving over 33,000 undiagnosed, with a possible 26,000 (80%) struggling in school as a direct result of CVI-related difficulties.

This pattern extends beyond Scotland. For instance, the Australian Childhood Visual Impairment Register recorded 904 participants, with CVI accounting for 15% (135 children) [18]. However, given that Australia has 5.74 million children, and Williams' [17] prevalence estimate suggests that 3.4% (195,000) could be affected, this means only 0.002% of children potentially affected by CVI have received a formal diagnosis. Not all countries have registries, but these findings strongly suggest that CVI is significantly underdiagnosed, highlighting the urgent need for improved screening, greater awareness, and enhanced diagnostic practices.

Research has identified two major environmental challenges affecting children with CVI: clutter and movement. Clutter includes visual distractions such as wall decorations, windows, work materials, and clothing patterns, which can make it difficult for children with CVI to visually map their environment [8,19,20]. Bennett [20] used virtual reality simulations to demonstrate that increasing clutter negatively impacted children with CVI when searching for objects or navigating hallways and McDowell [21] observed significant improvements in attention, focus, and behaviour when clutter was reduced in a special school setting.

Movement also presents challenges. Chandna [22] found that children with CVI struggle to process visual information in dynamic environments, as movement increases cognitive load, disrupting visual attention and spatial awareness. This can lead to anxiety, avoidance of visually overwhelming spaces, and difficulties in navigation and social interactions. Supportive strategies, such as reducing clutter, minimising movement in classrooms, and implementing structured routines, could help the estimated one undiagnosed child per school class struggling with learning and well-being [17,23].

This pilot study called **C.L.A.S.S.** (**C**erebral visual impairment **L**earning and **A**wareness for **S**chool **S**taff) was designed with four core objectives: to assess teachers' awareness of CVI across different educational settings, investigate current classroom practices related to CVI, compare the effectiveness of text-based versus virtual reality simulation video resources in increasing CVI awareness, and identify teachers' readiness to implement changes to support students with CVI.

Bite-size learning, sometimes called 'microlearning' is an increasingly popular approach in professional development, offering concise, focused resources designed to fit into busy schedules while maximising impact [24]. For teachers, this format provides an accessible way to learn about complex topics such as CVI without requiring significant time commitments. By delivering essential information in manageable portions, bite-size learning can enhance understanding and encourage practical application in the classroom. This study leverages the bite-size approach, using short video and text-based resources to educate teachers about CVI and its impact, aiming to improve awareness and support for children with CVI in schools.

This paper outlines the findings of the CLASS pilot study, offering insights into current awareness of CVI and strategies to improve awareness and in turn outcomes for children with CVI. By understanding teachers' perceptions and willingness to adapt their practices, this research aims to inform the development of effective interventions and resources to better support children with CVI in educational settings.

## Materials and methods

As groundwork for this study, the research team engaged in discussions with teachers and headteachers from their professional and personal networks to identify practical and effective approaches to support those affected by CVI. A consistent theme that emerged was the immense time pressure faced by teachers, underscoring the need for a format that was both concise and accessible. To address this, the study was designed as a brief, focused survey incorporating one of two educational media formats, video or text, designed to take less than five minutes to complete. This approach aimed to increase the likelihood of teacher participation while ensuring meaningful engagement.

The study evaluated four key areas: 1. Awareness of CVI, 2. Baseline of current teaching practices 3. Impact of different types of media on understanding, and 4. Changes teachers would be prepared to make in their classrooms. To support clarity, a flowchart summarising the overall study design, including recruitment, media allocation, survey components, and analysis, is included in S1 Appendix. The questionnaire (S2 Appendix) first asked teachers to specify the type of school they taught in, categorised as primary, secondary, special education, or other. If "other" was selected, participants were asked to provide further details about their area of teaching, such as specialist vision teachers. Teachers were then asked if they had heard of CVI to gauge baseline awareness across different teacher groups.

To establish a baseline of current practices, seven questions were developed to assess whether teachers reduced the number of items on classroom walls, wore plain, pattern-free clothing, wore similar plain clothing each day, sat or stood in consistent locations when teaching, used students' names when addressing them, displayed only one item at a time on boards or smart screens, and removed unnecessary items from smart screens, such as icons and wallpapers (Table 1). These responses provided a baseline of practices across different school types.

CVI experts developed the media used in the study, which included a short simulation video created using the University of St Andrew's CVI virtual reality programme, CVI-SIM [25]. The video explained what CVI is, how many children in schools are affected, the challenges they face, and actionable strategies teachers can adopt to support learning, attention,

**Table 1. Baseline teacher survey questions.** This table presents the seven questions asked of teachers to establish a baseline of their current practices. The same set of questions was asked again after exposure to the media to assess its impact on their understanding and approach.

| Question No | Question |
|---|---|
| 1 | Reduce the number of things on walls in classrooms. |
| 2 | Wear plain clothes without pattern. |
| 3 | Wear similar plain clothes each day. |
| 4 | Sit or stand in roughly the same place when teaching. |
| 5 | Use children's names when addressing them. |
| 6 | Only have one thing at a time on boards and smart screens. |
| 7 | Remove unnecessary items from the smart screen, for example icons and wallpaper. |

and well-being in class. The script (S2 Appendix) was designed to function independently as a standalone text, ensuring consistent messaging across formats. The video [26] was three minutes long, while the text version took approximately one and a half minutes to read.

Following the media, participants were asked the same seven questions again to measure any changes they might be prepared to make and to assess the effectiveness of the different media formats. Finally, teachers were prompted to reflect on the media and consider whether they could identify children with CVI among those they had taught and, if so, how many, with options ranging from zero to over fifty. While this measure was subjective, given that one teacher may have taught thousands of children and another only dozens, it provided insight into teachers' perceptions of CVI prevalence within their own classrooms.

The recruitment period for this study ran from 29 May 2023 to 10 July 2023, spanning six weeks.

Recruitment presented significant challenges, as contacting teachers required navigating different educational structures across the UK. In Scotland, local authority approval was required before a school could be contacted, except for independent schools, where the headteacher could be approached directly. Identifying the correct point of contact within local authorities was often difficult, with educational psychology departments frequently serving as a useful starting point. Some local authorities granted approval immediately, while others had lengthy internal ethics processes, and many did not respond at all.

In England and Wales, the process was different due to the prevalence of academies, which are no longer under local authority control, and many are part of multi-academy trusts (MATs). These administrative bodies oversee multiple schools, but there were no clear guidelines on how to approach them for research. Some MATs required direct contact with individual schools, while others centralised inquiries. Schools still under local authority control followed a process similar to Scotland, while independent schools could be approached directly.

Northern Ireland presented yet another distinct system, as there are no local education authorities. Instead, schools are categorised into different types, such as faith schools, free schools, and private schools, each requiring a different approach.

The study's ethics approval required strict adherence to the correct procedures when contacting schools. However, identifying the appropriate route often proved challenging. Even after obtaining approval from local authorities or administrative bodies, additional consent was required from the headteacher, and teacher participation remained voluntary. These steps were time-consuming and significantly impacted the research timeline, which was limited to six weeks.

The survey randomly allocated participants to one of two media formats: the video or text-based resource. The goal was to determine the most effective format for increasing teachers' understanding of CVI and guiding them toward implementing supportive changes in their classrooms. Additionally, the survey was designed to identify areas where teachers were more or less likely to make adjustments, providing valuable insights for future strategies to promote inclusive teaching practices.

The questionnaire used a 5-point Likert scale to measure participants' willingness to implement CVI-supportive strategies.

Participants' willingness to implement supportive practices in their classrooms increased after exposure to the media resources. Responses were scored using a 5-point Likert scale, with options ranging from "Definitely Not" (0) to "Definitely" (4), as follows:

- 0: Definitely Not

- 1: Unlikely

- 2: Not Sure

- 3: Likely

- 4: Definitely

An additional "Not Applicable" option was provided, and in the post-media questionnaire, an "Already Do" option was included to account for pre-existing practices.

The survey ran for six weeks, and data were exported for analysis. Descriptive statistics were used to summarise teacher awareness and reported practices across school types. Because the survey used ordinal Likert-scale responses, nonparametric tests were selected to assess changes in reported practices. Pre- and post-media responses were analysed using the Wilcoxon signed-rank test, applied independently to each media format. A chi-square test of independence was used to compare awareness levels and estimated prevalence of CVI across teacher groups. The chi-square analysis was performed using standard statistical tools, and the Wilcoxon analysis was supported by a custom script. Statistical significance was set at $p < .05$.

This study was open only to UK-based adult teachers, and no minors participated. Informed electronic consent was obtained through the survey platform Qualtrics before participation. Participants were first presented with a Participant Information Sheet outlining the study's purpose, data handling procedures, voluntary nature of participation, and withdrawal rights. Consent was explicitly obtained by requiring participants to check statements confirming their understanding before proceeding with the survey (see S2 Appendix). The University of St Andrews School of Medicine Ethics Committee approved this consent procedure under Approval Code: MD16936.

## Results

### Teacher participation by school type

A total of 143 teachers began the survey, and 111 completed the full questionnaire including exposure to the educational media. The statistical analysis is based on these 111 completed responses representing a variety of school settings across the UK. Participants included teachers from mainstream primary and secondary schools, special schools, and other educational contexts such as alternative provision and vision impairment support services. Table 2 summarises the distribution of participants by school type, including age ranges of pupils taught and contextual notes.

### Prevalence of awareness

A chi-square test of independence was performed to examine whether awareness of CVI significantly differed between mainstream (Primary + Secondary combined) and Special School teachers. The results were highly significant, $X^2(1, N = \text{total sample size}) = 135.01$, $p < .0001$, indicating a substantial disparity in CVI awareness between these groups. Specifically, mainstream teachers had significantly lower awareness of CVI compared to Special School teachers, reinforcing the need for targeted professional development in mainstream settings. Overall, the study found that 72% of participants had not heard of CVI or its related terms, highlighting a critical gap in awareness among teachers. The

**Table 2. Teacher participants by school type.** Distribution of participants in the CLASS pilot study, with contextual information on pupil age range and school setting. "Other" includes Qualified Teachers of Vision Impairment (QTVIs/TVIs) and teachers working in alternative provision.

**Teacher Participants by School Type**

| School Type | Number of Participants |
| --- | --- |
| Primary School Mainstream Teachers, working with children aged 11/12–18 | 43 |
| Secondary School Mainstream Teachers, working with children aged 4–11/12. | 44 |
| Special School Teachers, supporting students with additional needs. | 14 |
| 'Other' Teachers, including Qualified Teachers of Vision Impairment (QTVIs/TVIs) and educators in alternative settings. | 10 |

**Table 3. Awareness of Cerebral Visual Impairment (CVI) by School Type.** Percentage of participants in each school category who reported being unaware of CVI prior to engaging with the study materials. These figures reflect baseline awareness levels and form the basis of the chi-square analysis presented in the Results section. "Other Roles" includes specialist teachers such as Qualified Teachers of Vision Impairment (QTVIs).

**Awareness of CVI by School Type**

| School Type | Percentage Unaware of CVI |
| --- | --- |
| Primary School Teachers | 98% |
| Secondary School Teachers | 80% |
| Special School Teachers | 21% |
| Other Roles (e.g., QTVIs) | 0% |

findings emphasise the need for increased CVI education and training across all teaching groups, particularly in mainstream education. CVI awareness data were drawn from all 143 respondents, as this question was positioned early in the survey, before media exposure and survey drop-off. Table 3 presents the breakdown of awareness by school type (Figs 1 and 2).

These findings demonstrate a critical lack of awareness, particularly in mainstream settings, where children with CVI are underdiagnosed and unsupported.

## Changes in classroom practices

Pre-media, the average Likert score across seven questions was 2.6 (between "Not Sure" and "Likely"). Post-media, this score rose to 3.2 (between "Likely" and "Definitely").

These results suggest that concise educational media can significantly influence teachers' willingness to adopt evidence-based practices.

Table 4 presents the average pre- and post-media Likert scores across all teachers and both media conditions, illustrating overall changes in reported practices. To explore potential differences between teacher groups, Fig 3 organises the same results by teacher type, distinguishing between mainstream secondary, mainstream primary, special school, and other educators, before and after the media. This comparison allows for an analysis of whether the impact of the media varied across different educational settings.

## Findings on media influence and teacher responses

Question 5, which asked whether teachers use a child's name when addressing them, showed only a minimal increase from 3.8 to 3.9 on a 4-point scale. This indicates that the vast majority of teachers were already consistently using children's names, leaving little room for the media to influence this practice. Given that any potential impact could only have

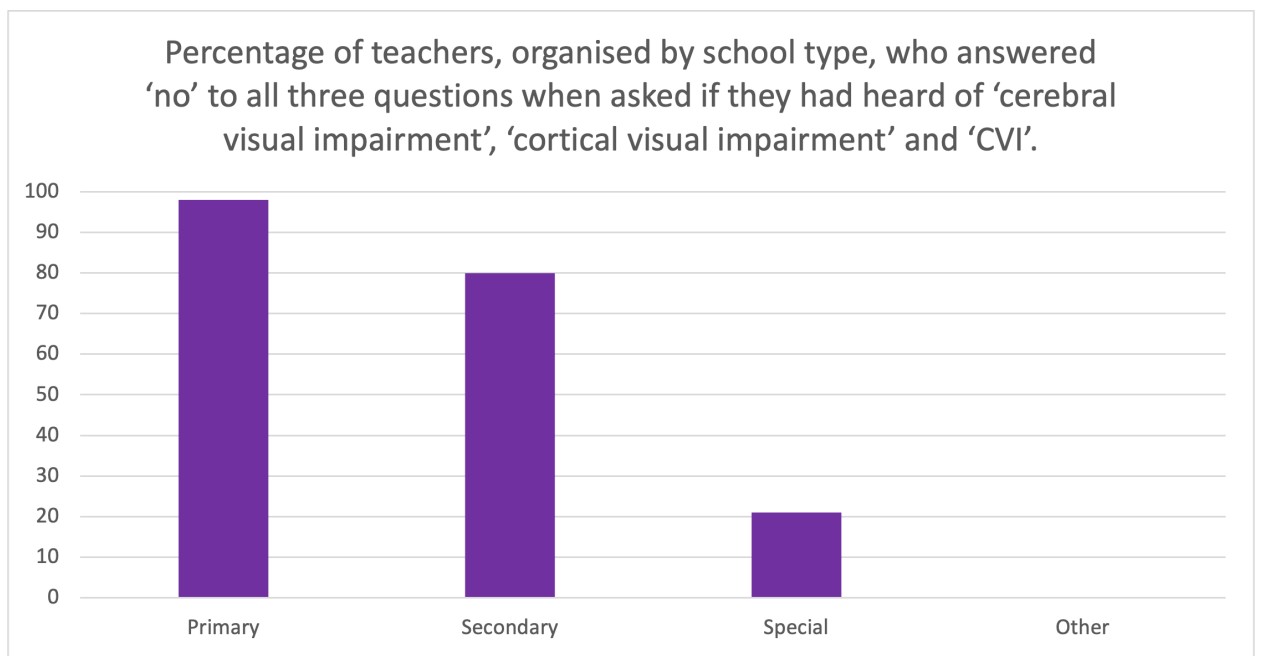

**Fig 1. Awareness of CVI among teachers by school type.** Percentage of teachers unaware of CVI, showing variation across school types.

been negative, an unlikely outcome that did not occur, Question 5 was excluded from further analysis of media impact. The decision to exclude this question ensures that the analysis focuses only on areas where meaningful change could be achieved.

By examining the changes in responses across teacher types and questions (Table 5), we can identify where media had the greatest influence and where it had the least impact. This highlights areas where investment in further interventions may be most effective, as well as areas where challenges may be more significant.

With the exception of the 'Other' category of teachers, many of whom were specialist vision teachers, the three main teacher groups (Primary, Secondary, and Special Education) demonstrated similar patterns in the changes they were willing to implement following media exposure (Fig 4). This consistency was particularly evident across primary and secondary mainstream teachers, despite likely differences in class sizes and student demographics. The alignment in their responses suggests that the strategies introduced through the media were broadly applicable across mainstream education settings, reinforcing their potential for widespread adoption.

However, subtle variations in the degree of change indicate that classroom context, student numbers, and existing teaching practices may influence how readily teachers adapt to new strategies.

## Media effectiveness

One item on the questionnaire, "Using the child's name when addressing them," was excluded from comparative analysis. This decision was made post hoc due to a ceiling effect: nearly all participants reported already doing this prior to exposure to the media, leaving little to no room for measurable change. As the primary aim of the questionnaire was to assess change in reported practice, this item could not meaningfully contribute to the analysis of media effectiveness. While the item was not identified as unsuitable during study design, its exclusion was necessary to maintain analytical validity. The finding itself is nonetheless valuable, suggesting that this particular inclusive practice is already well established among teachers.

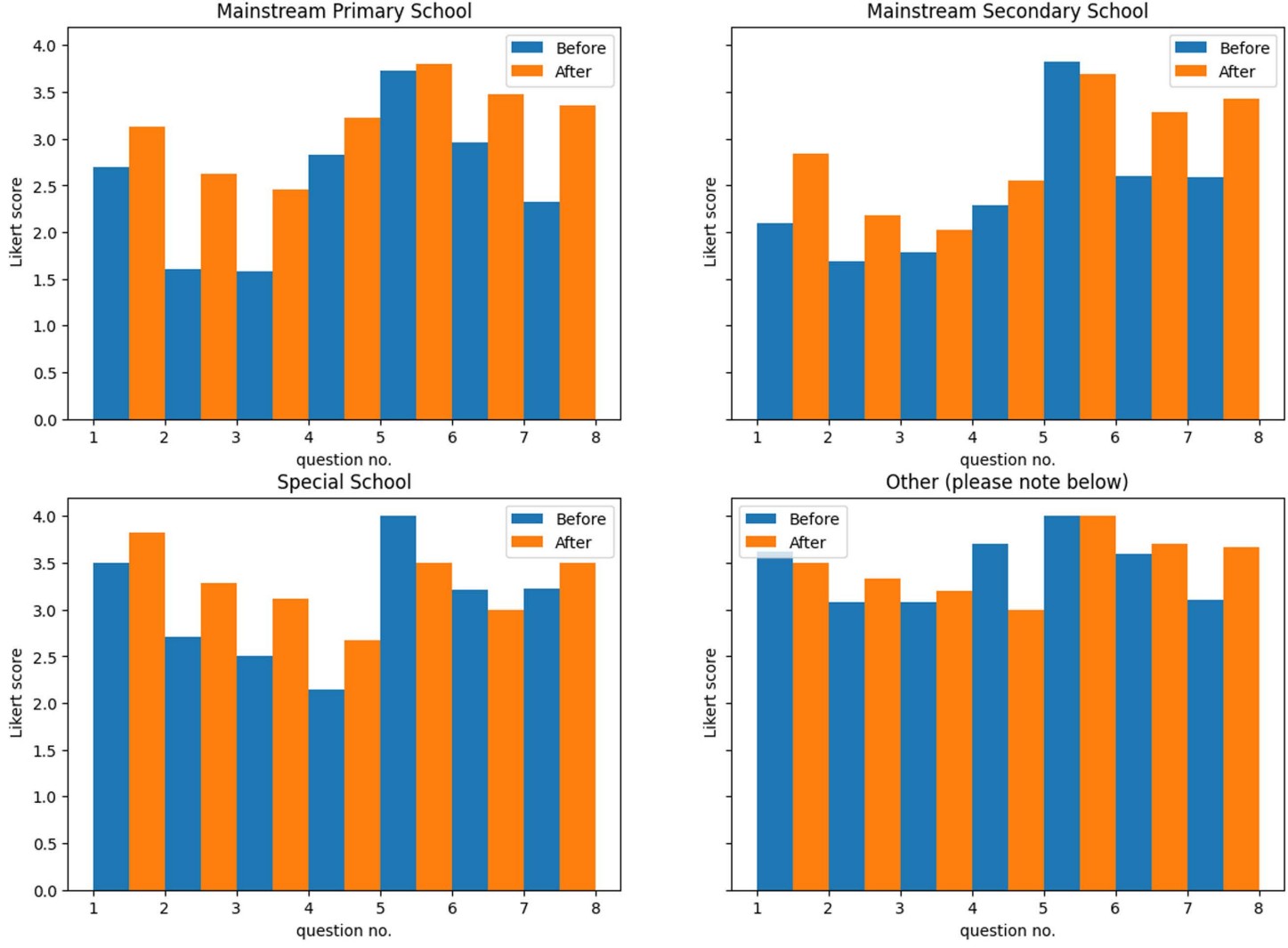

**Fig 2. Pre and post media likert scores by teacher type.** This figure consists of four bar charts, each representing a different teacher group (mainstream secondary, mainstream primary, special school, and other). Each chart displays the average Likert scores before and after viewing the educational media, allowing for a comparison of the media's impact across different educational settings.

A Wilcoxon signed-rank test was conducted to assess whether teachers' willingness to implement changes significantly increased after exposure to the media resources. The results indicated a highly significant increase for both media formats.

For the video format, teachers' willingness scores were significantly higher after exposure (median = 1.0) compared to before (median = 0.0), z = -8.90, p < .0001. Similarly, for the text format, willingness scores were significantly higher after exposure (median = 1.0) compared to before (median = 0.0), z = -12.91, p < .0001. These findings indicate a significant increase in reported willingness to implement CVI-supportive strategies following both media formats.

Both video and text-based resources effectively increased awareness and willingness to implement changes, with highly significant improvements (p < 0.0001) for both formats. The text format showed a slightly greater increase in average Likert scores, and the Wilcoxon signed-rank test revealed a larger statistical effect for text (z = -12.91) compared

**Table 4. Impact of media on teacher responses.** This table presents the average Likert scores for the seven baseline questions, both before and after exposure to the media. The final column shows the positive difference between the two scores, indicating the potential impact of the media on teacher understanding and practice.

| Pre & Post Media Average Likert Scores | | | | |
|---|---|---|---|---|
| | **Question** | **Pre** | **Post** | **Difference** |
| 1 | Reduce the number of things on walls in classrooms. | 2.5 | 3.2 | +0.7 |
| 2 | Wear plain clothes without pattern. | 1.9 | 2.8 | +0.9 |
| 3 | Wear similar plain clothes each day. | 1.9 | 2.7 | +0.8 |
| 4 | Sit or stand in roughly the same place when teaching. | 2.6 | 3.1 | +0.5 |
| 5 | Use children's names when addressing them. | 3.8 | 3.9 | +0.1 |
| 6 | Only have one thing at a time on boards and smart screens. | 2.8 | 3.4 | +0.6 |
| 7 | Remove unnecessary items from the smart screen, for example icons and wallpaper. | 2.6 | 3.5 | +0.9 |

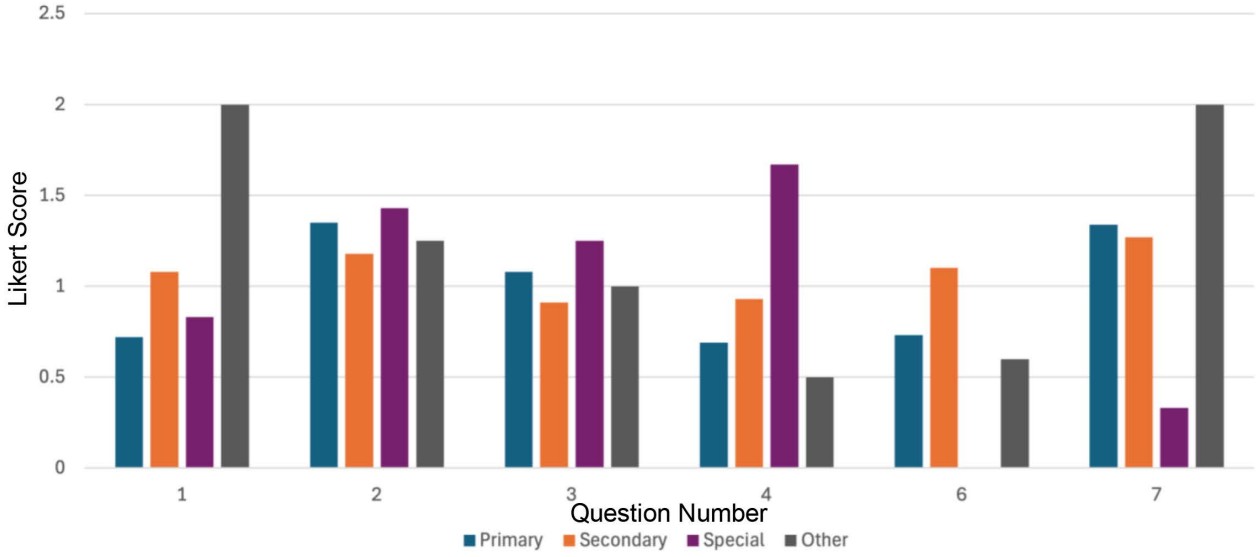

**Fig 3. Changes in teacher responses following media exposure.** This bar chart illustrates the positive change in teacher responses after media exposure, categorised by teacher type (Primary, Secondary, Special, and Other) and organised by question number (1, 2, 3, 4, 6, and 7). The height of the bars represents the magnitude of change, with higher values indicating a greater willingness to modify practice.

to video (z=-8.90). However, the video format was also highly effective, producing a similarly strong impact, with both formats achieving an identical median increase of 1.0. These results suggest that while text may have led to slightly larger shifts in rank-based scores, the video format remained a powerful and engaging tool for increasing teacher willingness to implement CVI-supportive strategies.

- Video Simulation: Pre-media Likert score of 2.8 (between "Not Sure" and "Likely"); post-media score of 3.3 (closer to "Likely").

- Text-Based Resource: Pre-media Likert score of 2.5 (just above "Not Sure"); post-media score of 3.2 (between "Likely" and "Definitely").

**Table 5. Changes in teacher responses following media exposure.** This table presents the positive change in teacher responses after media exposure, categorised by teacher type (Primary, Secondary, Special, and Other) and organised by question number (1, 2, 3, 4, 6, and 7). The values represent the magnitude of change, with higher numbers indicating a greater willingness to modify practice.

Changes in Teacher Responses Following Media Exposure by Teacher Type

| | Question Number | | | | | | Total |
|---|---|---|---|---|---|---|---|
| Teacher Type | 1 | 2 | 3 | 4 | 6 | 7 | |
| Primary | 0.72 | 1.35 | 1.08 | 0.69 | 0.73 | 1.34 | 5.91 |
| Secondary | 1.08 | 1.18 | 0.91 | 0.93 | 1.1 | 1.27 | 6.47 |
| Special | 0.83 | 1.43 | 1.25 | 1.67 | 0 | 0.33 | 5.51 |
| Other | 2 | 1.25 | 1 | 0.5 | 0.6 | 2 | 7.35 |

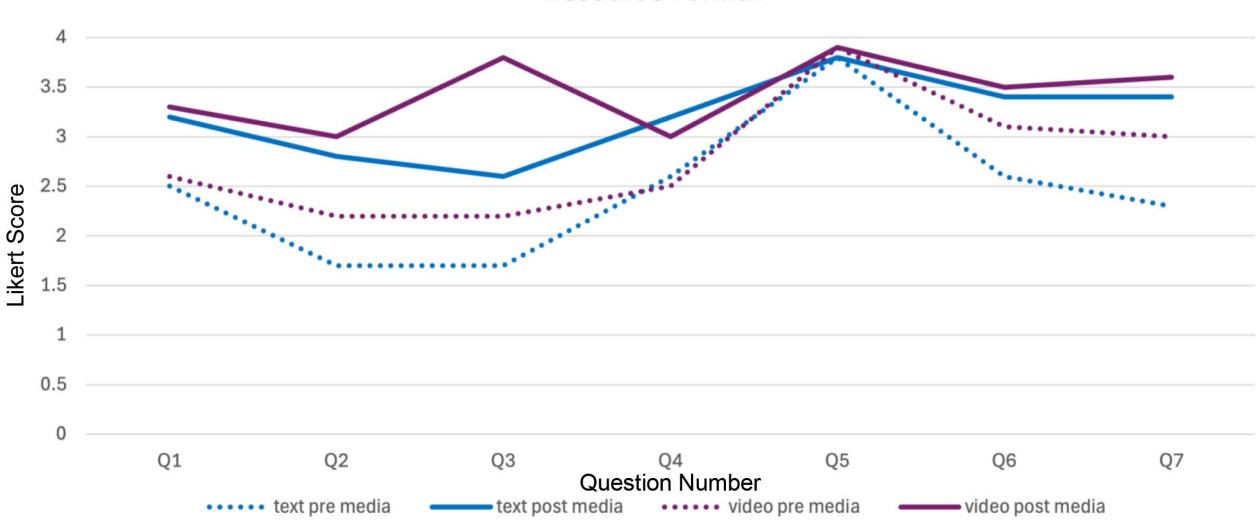

**Fig 4. Comparison of average Likert scores across seven classroom strategy questions (Q1–Q7) for teachers exposed to either text-based or video-based CVI resources.** Dotted lines represent pre-media scores; solid lines represent post-media scores. Scores range from 0 ("Definitely Not") to 4 ("Definitely"). Both formats resulted in increased willingness to implement CVI-supportive strategies, with text showing slightly larger improvements on several items.

These findings suggest that both media formats are valuable tools for raising awareness and promoting evidence-based adaptations. The highly significant P-value underscores their effectiveness in improving teaching practices, ultimately benefiting children with CVI.

### Identifying potential CVI cases

After exposure to media content, participants were asked to estimate the number of children in their classrooms who might have CVI. A chi-square test of independence was performed to examine whether the estimated number of students suspected to have CVI varied significantly between mainstream (Primary + Secondary combined) and Special School teachers. The test compared estimated CVI cases across six response categories: 0, 1, 2–5, 5–10, 10–50, and 50 + students. The results were not statistically significant, $X^2(5, N = \text{total sample size}) = 7.36, p = .195$, indicating that CVI estimates did not significantly differ between these teacher groups. This suggests that, despite differences in teaching environments,

both mainstream and special school teachers reported similar estimations of potential CVI cases. However, the number of children suspected to have CVI varied widely among respondents (see Table 6).

When the data are examined by teacher type, one might expect that secondary school teachers, who generally teach a larger number of students, would identify CVI in a greater number of children. However, most secondary school teachers reported suspecting CVI in fewer than ten students, a pattern that closely mirrors the responses of primary school teachers (see Fig 5). This observation, coupled with evidence that teachers are willing to implement CVI-positive changes based on increased awareness, is particularly noteworthy. It suggests that educators may be inclined to adjust some teaching strategies even when uncertainty exists regarding a student's CVI status, underscoring a strong commitment to making small, classroom-based adjustments within their control. Note, eight teachers did not respond to this question.

## Discussion

The discrepancy between the prevalence of CVI awareness among teachers and the likely prevalence of CVI in mainstream schools highlights an urgent need for improved awareness with education on practical strategies to improve the experience of those affected by CVI. From discussions with teachers while conducting groundwork for this study it is

**Table 6. Estimated number of students suspected to have CVI, following media exposure, by teacher type.**

| Teacher Type | Number of Children | | | | | |
| --- | --- | --- | --- | --- | --- | --- |
| | 0 | 1 | 2–5 | 5–10 | 10–50 | 50+ |
| Primary | 9 | 3 | 15 | 12 | 3 | 1 |
| Secondary | 7 | 4 | 19 | 7 | 5 | 1 |
| Special | 1 | 1 | 3 | 2 | 1 | 2 |
| Other | 0 | 0 | 1 | 3 | 3 | 0 |

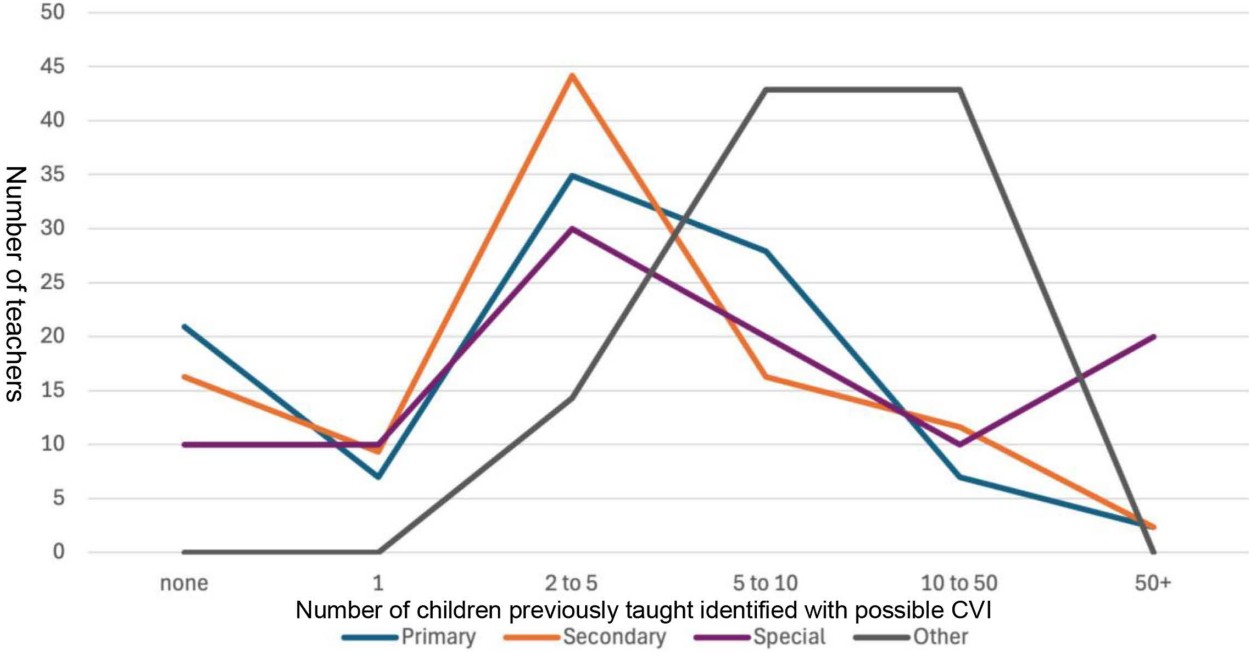

**Fig 5. Distribution of the number of students suspected to have CVI following media exposure, by teacher type.**

clear that teachers face enormous demands on their time, both in special and mainstream schools. They are required to navigate a diverse range of learning needs and styles, addressing abilities, difficulties, and disabilities that include autism, ADHD, dyslexia, dyspraxia, and other conditions such as anxiety, depression, and eating disorders.

Despite these significant demands, the study revealed an impressive willingness among teachers to make changes, even personal adjustments, to better support their students. Many of these changes fall within the teacher's immediate control and do not require broader school decisions or administrative approvals. For example, actions such as reducing wall clutter, wearing plain clothing, or simplifying smart screen displays are small yet impactful adjustments that teachers can implement autonomously within their classrooms.

This consistent willingness to adapt demonstrates that, while teachers' roles are increasingly challenging, they remain committed to helping where they can. These findings suggest that introducing small, manageable adjustments, especially those within the jurisdiction of individual teachers, may be the most effective way forward in equipping educators to support children with CVI, without adding undue strain to their already demanding roles.

Both media formats proved to be almost equally effective in increasing awareness and encouraging change. However, the virtual simulation video may be particularly beneficial for more educationally diverse populations, including parents, families, and carers. In the UK, the majority of adults function at an 11- to 14-year reading age [27], with one in six adults classified as functionally illiterate. This highlights the importance of accessible and inclusive resources, especially when addressing CVI, which has a higher prevalence in lower-income areas where educational attainment tends to be lower.

The virtual reality video, which simulates the visual experience of CVI, can be an effective tool for reaching these populations [28]. However, its effectiveness depends on ensuring the script, like the one designed for teachers in this study, is tailored to the needs of the target audience while maintaining expert quality and accuracy. This suggests that the use of virtual simulations could be expanded beyond the classroom to support families and communities, provided the content remains relevant, accessible, and evidence-based.

The resources used in this study were intentionally concise, three minutes for the video and under two minutes for the text, making them truly "bite-size." The results suggest that even short, focused resources can have a meaningful impact on awareness and willingness to make changes. This finding is particularly important when planning training and awareness initiatives for busy teachers, whose time is often constrained by the demands of their roles. Bite-size resources offer a practical and efficient way to deliver essential information, making professional development more accessible and achievable without adding significant burden to educators' schedules.

The consistency in responses, particularly across primary and secondary teachers, but also to a degree among special school teachers, despite differences in class sizes and student demographics, suggests that the strategies introduced through the media are broadly applicable across education settings. This adaptability is particularly valuable, as it indicates that a single set of resources can be effective for a wide range of educators (Figs 3, 4). The ability to apply these strategies across different teaching environments is further supported by the relatively uniform positive changes observed in teacher responses, reinforcing their potential for widespread use.

This adaptability improves efficiency by reducing the need for specialised training or multiple resource sets, saving time and effort for educators. Furthermore, it presents cost savings, as developing a single set of materials applicable across teacher groups minimises expenses associated with designing and implementing separate interventions.

Additionally, the scalability of these resources is enhanced by their broad applicability, ensuring effective integration into diverse school settings. The reduced need for extensive modifications lowers teacher training and professional development costs while also allowing for a more consistent implementation of strategies. This consistency makes it easier to monitor effectiveness and ensures equitable access to high-quality educational resources.

These findings highlight the potential for widespread adoption of the media-based strategies introduced in this study. However, variations in the magnitude of change across teacher groups, particularly in special schools and among the 'other' category, suggest that factors such as classroom context, student numbers, and existing teaching practices may

influence how readily teachers adapt to new strategies. Future research should explore these contextual influences in greater depth to refine the implementation of these resources and determine whether additional adaptations may be necessary for certain teaching environments.

Interpretation of these estimates should be approached with caution, given the considerable variation in classroom size and structure. For example, special school teachers may work intensively with just a few students, whereas mainstream secondary teachers often teach multiple classes across year groups. Additional factors such as career stage and pupil turnover may also influence these estimates. Given the modest sample size and scope of the pilot study, we chose not to present these findings as ratios or percentages, to avoid over-interpreting trends across groups. This approach will be re-evaluated in the next phase of research with a larger dataset.

Beyond questions of scale and format, this study also provides a foundation for influencing how CVI is approached within teacher professional development. As the first study to systematically measure teacher awareness of CVI, it highlights a fundamental barrier to inclusion: without basic recognition of the condition, particularly in mainstream schools, CVI is unlikely to be addressed in training programmes or classroom practice. While our research was time-limited by the six-week window of a summer scholarship scheme, it demonstrates that even short, accessible materials can spark awareness and willingness to change. The CLASS resources trialled here could be embedded within school-based training or local authority CPD frameworks. As professional development in the UK is often shaped at school or regional level, providing teachers with evidence-based, ready-to-use resources is essential. A further policy implication lies in the increasingly cluttered visual environments of classrooms. With mounting displays, labels, and decorations, visual overload is now common, and our findings support clearer guidance to reduce distractions and promote inclusive visual design, benefiting all learners, especially those with CVI.

While this study demonstrates the effectiveness of bite-size learning resources in improving teacher awareness and classroom adaptations for CVI, further research is needed to explore:

- Long-term retention and behavioural change – How sustained are these adaptations over time? Do teachers continue implementing them in practice?

- Impact on student outcomes – How do these changes in teacher behaviour influence the learning and well-being of children with CVI?

- Comparative effectiveness of different learning formats – Would interactive or scenario-based learning resources be more effective than video or text alone?

- Scaling and policy integration – How can these resources be incorporated into national teacher training programmes and education policies?

By addressing these questions, future research can help ensure that CVI-friendly teaching practices become an integral part of inclusive education, ultimately improving outcomes for children with CVI in both mainstream and specialist settings.

## Conclusion

This study set out to evaluate teacher awareness of CVI, assess current classroom practices, measure the impact of two different media formats, and determine the changes teachers would be prepared to implement following exposure to these resources. The findings provide valuable insights into each of these objectives.

A significant gap in CVI awareness was identified, particularly in mainstream schools, where 98% of primary school teachers and 80% of secondary school teachers had not heard of CVI. This highlights an urgent need for targeted professional development and accessible resources to bridge this knowledge gap. The baseline assessment of teaching practices revealed that while some teachers already employed CVI-friendly strategies, many did not consistently implement key adaptations such as reducing classroom clutter or simplifying smart screen content.

Both video and text-based resources significantly increased teacher awareness and willingness to adopt CVI-supportive practices, with highly significant improvements ($p < 0.0001$). While the text format showed slightly greater improvements in average Likert scores, the Wilcoxon test revealed a larger statistical effect for text ($z = -12.91$) compared to video ($z = -8.90$). However, the video format was also highly effective, achieving an identical median increase of 1.0, suggesting it had a strong and consistent impact across participants. These findings indicate that while both formats are effective, video remains a compelling option for delivering accessible, engaging educational resources to teachers.

A key finding was teachers' strong willingness to implement changes, even when uncertain whether any students in their classrooms had CVI but recognising it as a possibility. Notably, the most successful changes were those that were small, manageable, and within the teacher's direct control, such as reducing visual distractions, maintaining consistent positioning, and simplifying screen displays. These low-resource adaptations have the potential for significant impact, benefiting large numbers of children while remaining practical for teachers.

This study demonstrates the potential of bite-size learning resources to improve CVI awareness and drive meaningful change in classroom practices. By providing concise, accessible training, teachers can be equipped with effective strategies to support children with CVI without adding a significant burden to their workload. Future initiatives should focus on scaling these resources to reach a wider audience, including families and caregivers, while continuing to refine and evaluate their effectiveness in diverse educational settings.

## Limitations

This study has several limitations that should be considered when interpreting the findings.

The survey was conducted anonymously, which ensured honest responses but prevented follow-up questions or verification of self-reported changes in practice. Additionally, while the study design required participants assigned to the video condition to watch the full three-minute simulation before proceeding, those assigned to the text condition could technically advance without reading the full passage. However, an analysis of time spent on the text suggests that participants engaged with it adequately.

Another limitation relates to self-reported intentions rather than observed behavioural changes. While teachers expressed a strong willingness to implement CVI-friendly strategies, the study did not track whether these changes were sustained over time or effectively integrated into classroom practice.

The sample size was relatively small, which may limit the generalisability of the findings. However, the results were highly significant and consistent across teacher types, suggesting that the trends observed are meaningful. Larger-scale studies would be beneficial to confirm these patterns and further explore variations across different educational settings.

Despite these limitations, the findings provide valuable insights into teacher awareness of CVI and the effectiveness of concise, accessible learning materials in influencing classroom practices. Future research should explore longitudinal studies to assess the retention and real-world application of these adaptations over time.

## Supporting information

**S1 Appendix. Flowchart of Research Process.** A visual summary of the research process undertaken in the CLASS pilot study, including participant recruitment, media allocation, pre- and post-media questionnaires, and data analysis procedures.
(JPG)

**S2 Appendix. Complete survey content including participant information sheet, consent form, questionnaire, text-based media script, and debrief.**
(PDF)

**S3 Appendix. Raw data used in the statistical analysis for the CLASS Pilot Study.**
(XLSX)

## Acknowledgments

We gratefully acknowledge the support of the University of St Andrews Department of Computer Science, as well as the numerous students and staff who contributed to the ongoing work on cerebral visual impairment and the CVI simulation project (CVI-SIM). We also thank Patrick Millar for his invaluable assistance with data management and statistical analysis. Finally, we express our sincere gratitude to the teachers who generously took time from their busy schedules to complete our survey.

## Author contributions

**Conceptualization:** Aloka Jayasinghe, Helen St Clair Tracy, Andrew Blaikie.

**Data curation:** Aloka Jayasinghe.

**Formal analysis:** Aloka Jayasinghe, Helen St Clair Tracy, John Ravenscroft.

**Funding acquisition:** Aloka Jayasinghe.

**Investigation:** Aloka Jayasinghe, Helen St Clair Tracy, Andrew Blaikie.

**Methodology:** Aloka Jayasinghe, Helen St Clair Tracy, Andrew Blaikie.

**Project administration:** Aloka Jayasinghe, Helen St Clair Tracy, Andrew Blaikie.

**Resources:** Andrew Blaikie.

**Software:** Helen St Clair Tracy, Andrew Blaikie.

**Supervision:** Helen St Clair Tracy, Andrew Blaikie.

**Validation:** Helen St Clair Tracy, John Ravenscroft, Andrew Blaikie.

**Visualization:** Aloka Jayasinghe, Helen St Clair Tracy, Andrew Blaikie.

**Writing – original draft:** Helen St Clair Tracy.

**Writing – review & editing:** Aloka Jayasinghe, Helen St Clair Tracy, John Ravenscroft, Andrew Blaikie.

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
