## [Decision Letter · Decision Letter 0]

31 Mar 2025

PONE-D-25-11648The CLASS (Cerebral visual impairment Learning and Awareness for School Staff) Pilot Study: An evaluation of the awareness of CVI amongst teachers and comparative evaluation of two different educational resources on understanding.PLOS ONE

Dear Dr. St Clair Tracy,

Thank you for submitting your manuscript to PLOS ONE. After careful consideration, we feel that it has merit but does not fully meet PLOS ONE’s publication criteria as it currently stands. Therefore, we invite you to submit a revised version of the manuscript that addresses the points raised during the review process.

We look forward to receiving your revised manuscript.

Kind regards,

Jordan Llego, PhD ELM, D. Hon. Ex., PhDN, RN

Academic Editor

PLOS ONE

2. In the online submission form you indicate that your data is not available for proprietary reasons and have provided a contact point for accessing this data. Please note that your current contact point is a co-author on this manuscript. According to our Data Policy, the contact point must not be an author on the manuscript and must be an institutional contact, ideally not an individual. Please revise your data statement to a non-author institutional point of contact, such as a data access or ethics committee, and send this to us via return email. Please also include contact information for the third party organization, and please include the full citation of where the data can be found.

Additional Editor Comments:

Thank you for submitting your manuscript titled "The CLASS (Cerebral Visual Impairment Learning and Awareness for School Staff) Pilot Study: An Evaluation of the Awareness of CVI Amongst Teachers and Comparative Evaluation of Two Different Educational Resources on Understanding" to PLOS ONE. We appreciate your effort, and after reviewing the comments from the reviewers, we are pleased to inform you that both have recommended a minor revision. Your study addresses a crucial gap in awareness and education surrounding Cerebral Visual Impairment (CVI), demonstrating commendable clarity and rigor. Using bitesized educational interventions for teacher awareness is particularly timely and valuable. However, several points must be addressed to enhance the manuscript's clarity, structure, and rigor. Please find the comments of the reviewers below, and here are my recommendations as well:

First, ensuring consistent terminology throughout the text is important, especially regarding acronyms such as CVI and phrases like "bitesize learning" and "media format." Defining these acronyms upon first mention in the main text will aid in understanding. Additionally, some paragraphs in the "Results" section are lengthy; consider breaking these down into more digestible segments and adding subheadings where applicable to improve readability.

In the discussion section, expanding on how your findings can be integrated into existing teacher professional development programs would be beneficial. Providing concrete recommendations or policy suggestions will strengthen the practical implications of your research. Furthermore, attention should be given to the visual quality of figures; specifically, Figures 4 and 5 could be improved with better resolution and visual clarity. Ensuring that fonts are legible and that color schemes are accessible to all viewers, including color-blind viewers, is essential. Finally, please cross-check all citations to ensure they align with the reference list and adhere to the formatting guidelines specified by PLOS.

Thank you for your hard work, and we look forward to your revised manuscript.

Reviewers' comments:

Reviewer's Responses to Questions

**Comments to the Author**

1. Is the manuscript technically sound, and do the data support the conclusions?

Reviewer #1: Yes

Reviewer #2: Yes

2. Has the statistical analysis been performed appropriately and rigorously? 

Reviewer #1: No

Reviewer #2: Yes

3. Have the authors made all data underlying the findings in their manuscript fully available?

Reviewer #1: Yes

Reviewer #2: No

4. Is the manuscript presented in an intelligible fashion and written in standard English?

Reviewer #1: Yes

Reviewer #2: Yes

5. Review Comments to the Author

Reviewer #1: Introduction: I recommend that the introduction section provides a more detailed overview of teachers' current understanding of Cerebral Visual Impairment (CVI) and emphasizes the important social significance of enhancing teachers' awareness of this condition.

Research Population Selection: To ensure that this study is representative, could you further elaborate on the criteria and methods used for selecting the research population? Additionally, please clarify the approach taken to choose the participating schools and how the sample size was calculated.

Research Process Illustration: I suggest creating a flowchart to clearly illustrate the research process, which would enhance the clarity of your methodology.

Reviewer #2: I wish to commend the authors for putting in great effort to put this work together in the midst of the challenges they faced. CVI is an often-overlooked condition and often overlooked condition which negatively impacts many lives. Studies like this are essential in putting a spotlight on the condition. Below are some comments that may help improve the current state of the manuscript

Comment 1

Can the authors introduce a section on analysis in the methods?

Comment 2

Even though the authors provide details of ethical processes they had to undergo, they do not provide any ethics approval number

Comment 3

In line 185 to 189, the authors try to present the characteristics of participants, this will have better been presented in a table format, which will have given room to provide more details about them, e.g. years of teaching, gender etc.

Comment 4

Line 191 to 200, the authors describe the level of awareness and provide bulleted statistics below it. These statistics will have been better presented as a table, since that will give more details and will help in appreciation of the chi-square analysis.

Comment 5

The authors can move the description of the scoring of the questionnaire (line 217 to 227) to methods

The removal of item 5 before from subsequent analysis seems a little out of place. All items in the questionnaire contribute to the measure of a specific quantity, to remove an item there must be enough evidence that will suggest its redundant. The evidence presented of a marginal pre and post score is not a known means of eliminating an item from a questionnaire. The authors may provide a better statistical justification for its removal

Comment 6

The authors often discuss the findings in the results section. It will be helpful to the reader if this is avoided and all such statements moved to the discussion section. An example is the discussion in line 305 to 308

Comment 7

The authors have indicated the bias that can be introduced by comparing the number of potential CVI cases among the various categories of teachers. Presenting the results as percentages or ratios may help to reduce the impact of the different class sizes.

Comment 8

The authors state as one of their objectives “, compare the effectiveness of text-based versus immersive virtual reality simulation video resources in increasing CVI awareness” however there were no descriptive results that demonstrated this, except for the chi-square results presented. I will suggest the authors present results that segregates responses based on media type.

Comment 9

Figure 4 and 5 have no x-axis and y-axis labelling

Comment 10

The information provided in table 2 is the same as in figure 2. The authors may remove one of them.

6. PLOS authors have the option to publish the peer review history of their article (what does this mean? ). If published, this will include your full peer review and any attached files.

**Do you want your identity to be public for this peer review?** For information about this choice, including consent withdrawal, please see our Privacy Policy .

Reviewer #1: No

Reviewer #2: **Yes: ** Carl Halladay Abraham

---

## [Author Response · Author response to Decision Letter 0]

28 Apr 2025

Journal Requirements

1. Formatting and File Naming:

o The manuscript has been checked against PLOS ONE formatting templates.

o Files have been named according to guidelines: Manuscript, Revised Manuscript with Track Changes, and Response to Reviewers.

2. Ethics Statement Placement:

o The ethics statement has been moved to the end of the Methods section and includes the approval code: MD16936.

3. Data Availability Statement:

o A revised statement has been included, directing requests to the University of St Andrews School of Medicine Ethics Committee (medresearch@st-andrews.ac.uk) in compliance with PLOS data sharing policy.

4. Supporting Information:

o All references to the Supporting Information have been updated to "S1 Appendix" & S2 Appendix”.

o A caption has been added at the end of the manuscript under a "Supporting Information" heading.

o There are now two appendices: S1, a newly added flowchart illustrating the research process (as recommended by Reviewer 1), and S2, which contains the questionnaire, consent form, participant information sheet, text-based media script, and debrief.

5. References:

o References 1 and 2 have been reformatted to align with Vancouver style.

o Several online/media citations (References 25–28) have been updated for clarity and consistency.

o Reference 28 has been updated to the published version of the iLRN 2024 conference paper.

6. Additional Note Regarding Figures:

o All revised figure files (Figures 1–5) have been uploaded to the Preflight Analysis and Conversion Engine (PACE) at https://pacev2.apexcovantage.com/ in accordance with PLOS ONE submission guidelines. Each figure was reviewed and amended to meet the required specifications, then reuploaded via PACE. We confirm that all figures now comply with journal standards and have been submitted through the appropriate platform. Please let us know if any further action is required.

Editor

Editor Comment

First, ensuring consistent terminology throughout the text is important, especially regarding acronyms such as CVI and phrases like "bitesize learning" and "media format." Defining these acronyms upon first mention in the main text will aid in understanding. Additionally, some paragraphs in the "Results" section are lengthy; consider breaking these down into more digestible segments and adding subheadings where applicable to improve readability.

Author Response:

Thank you for these helpful suggestions. We have made the following changes in response:

• CVI Definition: The term CVI is defined on first mention in the Introduction.

• Terminology Consistency:

o “bite-size” is used consistently throughout the manuscript.

o “media format” has been used in place of “media type” for consistency.

o The term “immersive” has been removed or clarified where appropriate to avoid potential bias.

• Paragraph Length: The Results section has been reviewed and revised for readability; several long paragraphs have been broken into shorter, more digestible sections.

Continuity and Presentation Order:

To improve consistency and reader clarity, we have standardised the order of school types throughout the manuscript, tables, and figures. References to participant groups now consistently follow the order: Primary, Secondary, Special, Other.

Editor Comment:

In the discussion section, expanding on how your findings can be integrated into existing teacher professional development programs would be beneficial. Providing concrete recommendations or policy suggestions will strengthen the practical implications of your research.

Author Response:

Thank you for this helpful suggestion. We have added a new paragraph to the Discussion section, highlighting how this study could inform the development of professional development resources for teachers. As the first study to systematically measure teacher awareness of CVI, our findings demonstrate an urgent need to raise awareness—particularly in mainstream schools, where CVI remains largely unknown. Without broader awareness, the inclusion of CVI in formal teacher training programmes remains unlikely. While our study was limited to a six-week data collection window due to the constraints of a summer research scholarship, it serves as a strong foundation for further work.

The CLASS materials tested in this study are themselves effective awareness-raising tools and could be embedded into school-based training sessions or local authority CPD frameworks. We have also acknowledged that teacher training in the UK is often school- or authority-led, highlighting the importance of influencing practice through accessible, evidence-based resources. Additionally, we point to the issue of increasingly cluttered classroom environments, which poses particular challenges for children with CVI. Based on our findings, we propose practical, policy-relevant strategies such as encouraging the reduction of unnecessary visual stimuli in classrooms. These points are now addressed in the revised Discussion section.

Manuscript Addition

Beyond questions of scale and format, this study also provides a foundation for influencing how CVI is approached within teacher professional development. As the first study to systematically measure teacher awareness of CVI, it highlights a fundamental barrier to inclusion: without basic recognition of the condition, particularly in mainstream schools, CVI is unlikely to be addressed in training programmes or classroom practice. While our research was time-limited by the six-week window of a summer scholarship scheme, it demonstrates that even short, accessible materials can spark awareness and willingness to change. The CLASS resources trialled here could be embedded within school-based training or local authority CPD frameworks. As professional development in the UK is often shaped at school or regional level, providing teachers with evidence-based, ready-to-use resources is essential. A further policy implication lies in the increasingly cluttered visual environments of classrooms. With mounting displays, labels, and decorations, visual overload is now common, and our findings support clearer guidance to reduce distractions and promote inclusive visual design—benefiting all learners, especially those with CVI.

Reviewer 1

Reviewer 1 Comment:

I recommend that the introduction section provides a more detailed overview of teachers' current understanding of Cerebral Visual Impairment (CVI) and emphasizes the important social significance of enhancing teachers' awareness of this condition.

Author Response:

Thank you for this comment. We have clarified in the Introduction that, to our knowledge, no prior research has systematically evaluated teacher awareness of CVI—our study is the first to address this gap. The very low levels of awareness identified (e.g. 98% of primary teachers had never heard of CVI) support this assertion and highlight the urgent need for improved education and training. We have also expanded our discussion of the broader social significance of teacher awareness, noting that undiagnosed CVI can contribute to misunderstanding, stigma, and exclusion in classroom environments. Without informed recognition, children with CVI may be misperceived as inattentive, poorly behaved, or less capable, leading to inappropriate responses and missed opportunities for support. These points are now more clearly articulated in the revised Introduction.

Manuscript Addition

Low awareness of CVI among teachers can have important social implications. Without a clear understanding of the condition, children’s behaviours may be misinterpreted, for example as inattentiveness or disengagement, rather than recognised as indicators of unmet access needs. These misunderstandings can unintentionally lead to reduced support, missed learning opportunities, and social exclusion, particularly in mainstream education settings, where teachers are already navigating a wide range of learning needs.

Reviewer 1 Comment:

To ensure that this study is representative, could you further elaborate on the criteria and methods used for selecting the research population? Additionally, please clarify the approach taken to choose the participating schools and how the sample size was calculated.

Author Response:

Thank you for this comment. We aimed to recruit a minimum of 382 teachers across the UK, based on a target of 95% confidence with a 5% margin of error from the overall teacher population of approximately 650,000. Our recruitment strategy sought to include primary, secondary, and special school teachers from all four UK nations. However, as described in the manuscript, we encountered substantial challenges navigating varied recruitment policies across local authorities, academies, and independent schools. These issues significantly affected participation and are discussed in detail in the Methods section, as we believe they reflect important structural barriers to conducting educational research in the UK. While our final sample fell below the original target, it included a broad range of teachers from diverse school types and settings. This study forms part of a larger, ongoing programme of work that is now extending recruitment across the UK.

Reviewer 1 Comment:

I suggest creating a flowchart to clearly illustrate the research process, which would enhance the clarity of your methodology.

Author Response:

Thank you for this excellent suggestion. We have created a flowchart illustrating the key stages of the research process, including participant recruitment, random allocation to media conditions, questionnaire structure, and data analysis. This has been included as S2 Appendix in the revised submission and is referenced in the Methods section to support clarity and transparency.

Reviewer 2

Reviewer 2 Comment 1:

Can the authors introduce a section on analysis in the methods?

Author Response:

Thank you very much for this helpful suggestion. We agree that a clear explanation of the analytical approach adds value and improves transparency for the reader. We have now added a paragraph at the end of the Materials and Methods section describing the statistical tests used (Wilcoxon signed-rank and chi-square), the rationale for using nonparametric methods for Likert-scale data, and the general analytical procedure. We greatly appreciate this comment and hope the addition strengthens the clarity of our methodology.

Manuscript Addition:

The survey ran for six weeks, and data were exported for analysis. Descriptive statistics were used to summarise teacher awareness and reported practices across school types. Because the survey used ordinal Likert-scale responses, nonparametric tests were selected to assess changes in reported practices. Pre- and post-media responses were analysed using the Wilcoxon signed-rank test, applied independently to each media format. A chi-square test of independence was used to compare awareness levels and estimated prevalence of CVI across teacher groups. The chi-square analysis was performed using standard statistical tools, and the Wilcoxon analysis was supported by a custom script. Statistical significance was set at p < .05.

Reviewer 2 Comment 2:

Even though the authors provide details of ethical processes they had to undergo, they do not provide any ethics approval number.

Author Response:

Thank you for highlighting this, and we also thank the editor for noting it in their summary. We apologise for the confusion — the ethics approval number was included in the original manuscript, but the full ethics statement had been placed at the end of the manuscript rather than within the Materials and Methods section, as required by PLOS ONE. We have now repositioned the complete statement in the correct location, and it includes the approval code (MD16936) as specified. We appreciate your careful reading and hope this resolves the issue.

Reviewer 2 Comment 3:

In line 185 to 189, the authors try to present the characteristics of participants. This would have been better presented in a table format, which would have given room to provide more details about them, e.g. years of teaching, gender etc.

Author Response:

Thank you for this helpful suggestion. We agree that a table improves clarity, particularly at the start of the Results section. While we did not collect data on gender, years of teaching experience, or region, we have reformatted the participant characteristics into Table 1, which now summarises school type, pupil age range, and educational context. We believe this improves readability and appreciate your recommendation.

Reviewer 2 Comment 4:

Line 191 to 200, the authors describe the level of awareness and provide bulleted statistics below it. These statistics would have been better presented as a table, since that will give more detail and will help in appreciation of the chi-square analysis.

Author Response:

Thank you for another helpful and thoughtful suggestion. We have replaced the bulleted list with Table 3, which presents CVI awareness levels by school type. We agree that the table format improves clarity and allows for better appreciation of the chi-square analysis referenced in the surrounding text. We’re grateful for your attention to presentation.

Reviewer 2 Comment 5:

The authors can move the description of the scoring of the questionnaire (line 217 to 227) to Methods.

Author Response:

Thank you for this suggestion. We have now moved the description of the Likert-scale scoring and response options to the Materials and Methods section, as recommended.

Reviewer 2 Comment 6:

The removal of item 5 before from subsequent analysis seems a little out of place. All items in the questionnaire contribute to the measure of a specific quantity; to remove an item there must be enough evidence that will suggest its redundancy. The evidence presented of a marginal pre and post score is not a known means of eliminating an item from a questionnaire. The authors may provide a better statistical justification for its removal.

Author Response:

Thank you for raising this important point. We agree that removing an item from analysis requires clear justification, and we have now strengthened our explanation accordingly in the Results section.

The primary aim of the questionnaire was to assess change in teachers’ reported willingness to implement CVI-supportive practices, following exposure to media resources. Item 5 (“Using the child’s name when addressing them”) was excluded from comparative analysis due to a clear ceiling effect: nearly all participants reported already doing this before viewing either media format. As a result, there was no meaningful opportunity to observe change or media impact. This was not evident at the point of study design, but became apparent upon initial analysis.

The item was not removed because it was redundant in general, but because it did not serve the analytic purpose of evaluating change, which was central to our use of the Wilcoxon test. We have now included a paragraph in the Results section that outlines this rationale more clearly. We also note that the finding itself is useful, suggesting that this particular inclusive practice is already well embedded. We appreciate your thoughtful feedback and hope this clarification addresses your concern.

Manuscript Addition

One item on the questionnaire, ‘Using the child’s name when addressing them,’ was excluded from comparative analysis. This decision was made post hoc due to a ceiling effect: nearly all participants reported already doing this prior to exposure to the media, leaving little to no room for measurable change. As the primary aim of the questionnaire was to assess change in reported practice, this item could not meaningfully contribute to the analysis of media effectiveness. While the item was not identified as unsuitable during study design, its exclusion was necessary to maintain analytical validity. The finding itself is nonetheless valuable, suggesting that this particular inclusive practice is already well established among teachers.

Reviewer 2 Comment 7:

The authors often discuss the findings in the Results section. It will be helpful to the reader if this is avoided and all such statements moved to the Discussion section. An

---

## [Editor Report · Decision Letter 1]

4 May 2025

The CLASS (Cerebral visual impairment Learning and Awareness for School Staff) Pilot Study: An evaluation of the awareness of CVI amongst teachers and comparative evaluation of two different educational resources on understanding.

PONE-D-25-11648R1

Dear Dr. St Clair Tracy,

We’re pleased to inform you that your manuscript has been judged scientifically suitable for publication and will be formally accepted for publication once it meets all outstanding technical requirements.

Kind regards,

Jordan Llego, PhD ELM, D. Hon. Ex., PhDN, RN

Academic Editor

PLOS ONE

Additional Editor Comments (optional):

Your study presents important and timely insights into the awareness of Cerebral Visual Impairment (CVI) among educators and the comparative impact of two educational interventions. The clarity of your methodology, the strength of your data analysis, and the practical implications for inclusive teaching practices were all commended by reviewers. The thoughtful integration of reviewer feedback has strengthened the manuscript considerably, particularly in its expanded discussion on policy relevance and professional development applications.

We are particularly appreciative of the care taken to clarify methodological details, justify analytical choices (e.g., item exclusion due to ceiling effect), and improve data presentation through new figures and tables. Your commitment to transparency and educational impact aligns well with the values of PLOS ONE.
---

## [Editor Report · Acceptance letter]

PONE-D-25-11648R1

PLOS ONE

Dear Dr. St Clair Tracy,

I'm pleased to inform you that your manuscript has been deemed suitable for publication in PLOS ONE. Congratulations! Your manuscript is now being handed over to our production team.

Kind regards,

on behalf of

Dr. Jordan Llego

Academic Editor

PLOS ONE